# Treatment of Winery Wastewater by Combined Almond Skin Coagulant and Sulfate Radicals: Assessment of ${\mathrm{HSO}}_{5}^{-}$ Activators

**DOI:** 10.3390/ijerph20032486

**Published:** 2023-01-30

**Authors:** Nuno Jorge, Ana R. Teixeira, Lisete Fernandes, Sílvia Afonso, Ivo Oliveira, Berta Gonçalves, Marco S. Lucas, José A. Peres

**Affiliations:** 1Escuela Internacional de Doctorado (EIDO), Campus da Auga, Campus Universitário de Ourense, Universidade de Vigo, As Lagoas, 32004 Ourense, Spain; 2Centro de Química de Vila Real (CQVR), Departamento de Química, Universidade de Trás-os-Montes e Alto Douro (UTAD), Quinta de Prados, 5000-801 Vila Real, Portugal; 3Centre for the Research and Technology of Agro-Environmental and Biological Sciences (CITAB), Universidade de Trás-os-Montes e Alto Douro (UTAD), Quinta de Prados, 5000-801 Vila Real, Portugal

**Keywords:** almond skin extract, Box–Behnken, coagulation–flocculation, response surface methodology, sludge valorization, SR-AOPs, wine production

## Abstract

The large production of wine and almonds leads to the generation of sub-products, such as winery wastewater (WW) and almond skin. WW is characterized by its high content of recalcitrant organic matter (biodegradability index < 0.30). Therefore, the aim of this work was to (1) apply the coagulation–flocculation–decantation (CFD) process with an organic coagulant based on almond skin extract (ASE), (2) treat the organic recalcitrant matter through sulfate radical advanced oxidation processes (SR-AOPs) and (3) evaluate the efficiency of combined CFD with UV-A, UV-C and ultrasound (US) reactors. The CFD process was applied with variation in the ASE concentration vs. pH, with results showing a chemical oxygen demand (COD) removal of 61.2% (0.5 g/L ASE, pH = 3.0). After CFD, the germination index (GI) of cucumber and corn seeds was ≥80%; thus, the sludge can be recycled as fertilizer. The SR-AOP initial conditions were achieved by the application of a Box–Behnken response surface methodology, which described the relationship between three independent variables (peroxymonosulfate (PMS) concentration, cobalt (Co^2+^) concentration and UV-A radiation intensity). Afterwards, the SR-AOPs were optimized by varying the pH, temperature, catalyst type and reagent addition manner. With the application of CFD as a pre-treatment followed by SR-AOP under optimal conditions (pH = 6.0, [PMS] = 5.88 mM, [Co^2+^] = 5 mM, T = 343 K, reaction time 240 min), the COD removal increased to 85.9, 82.6 and 80.2%, respectively, for UV-A, UV-C and US reactors. All treated wastewater met the Portuguese legislation for discharge in a municipal sewage network (COD ≤ 1000 mg O_2_/L). As a final remark, the combination of CFD with SR-AOPs is a sustainable, safe and clean strategy for WW treatment and subproduct valorization.

## 1. Introduction

The agro-industries account for 70% of all freshwater withdrawals worldwide, with the wine industry covering three sectors of the economy: agriculture, manufacturing and trade [1]. Due to the consumer demand for quality wines, washing and disinfection operations are necessary, and it is estimated that a winery produces about 1.3 to 1.5 kg of residue per liter of wine produced, 75% of which is winery wastewater (WW). Without a proper treatment process, the WW environmental impact is enormous due to the pollution of the water, degradation of the soil, damage to the vegetation, release of odors into the air, eutrophication of water resources, and consumption of oxygen from the rivers and lakes, leading to the suffocation of aquatic and amphibious life [2,3].

The almond (*Prunus dulcis* (Miller) D.A. Webb) is a member of the genus *Prunus*. The widespread distribution of the almond is related to its low chilling requirement, which makes almonds very suitable for regions with mild winters and dry, hot summers. World almond production has increased from 3.1 million to 4.1 million metric tons from 2011 to 2020 [4]. The almond industrial processes lead to the generation of large amounts of waste related to the removal of shells and skins [5]. Therefore, in this work, it is proposed to valorize the almond skin as a coagulant to treat WW via the CFD process.

The CFD process is a straightforward and economical process that has the capability to remove organic and inorganic substances and colloidal particles, depending on operational conditions, coagulant type and wastewater characteristics [6]. Although almond skin has never been used before in WW treatment, it has been observed that the extraction of active compounds from plants achieved significant results in wastewater treatment [7,8] and the sludge could be recycled for fertilization [9].

Considering the chemical composition of WW and the presence of recalcitrant organic matter in WW, one possible treatment strategy could be the use of advanced oxidation processes (AOPs). Among AOPs, hydroxyl-based AOPs (HR-AOPs) or sulfate-based AOPs (SR-AOPs) can be applied. In AOPs, hydroxyl radicals (${\mathrm{HO}}^{•}$) are generated by a number of processes. These radicals have an oxidation potential of 2.8 V, they are non-selective, reacting with the pollutants and oxidizing them to CO_2_, H_2_O and partially oxidized species [10,11].

In the past years, persulfates, such as peroxymonosulfate (PMS, ${\mathrm{HSO}}_{5}^{-}$) and peroxydisulfate (PDS, ${S_{2}O}_{8}^{2-}$), have attracted increasing attention because they are much more stable than hydrogen peroxide [12]. In addition, the sulfate radicals (${\mathrm{SO}}_{4}^{•-}$) produced by the activation of PMS and PDS (1) are more selective than ${\mathrm{HO}}^{•}$ radicals for the oxidation of compounds with carbon-carbon double bonds and benzene rings, and (2) have a higher oxidation potential than ${\mathrm{HO}}^{•}$ radicals ($E^{0}$ = 2.5–3.1 V) [13,14]. PMS is the active ingredient of triple potassium salt (KHSO_5_•0.5KHSO_4_•0.5K_2_SO_4_): with an oxidation potential of 1.82 V, it is stable at ambient temperature, easy to handle and the bond energy is estimated to be in the range of 140–213.3 kJ/mol [15]. To generate the ${\mathrm{SO}}_{4}^{•-}$ radicals, the PMS can be activated by metal catalysts, heat, UV, visible light, ultrasound (US), alkali and photo-catalytic activation ($h^{+}/e^{-}$) (Equations (1)–(6)) [16].
(1)$$M^{n+}+{\mathrm{HSO}}_{5}^{-}\;\to \;M^{(n+1)+}+{\mathrm{SO}}_{4}^{•-}+{\mathrm{HO}}^{-}$$
(2)$$M^{(n+1)+}+{\mathrm{HSO}}_{5}^{-}\;\to \;{\mathrm{SO}}_{5}^{•-}+M^{n+}+H^{+}$$
(3)$${\mathrm{HSO}}_{5}^{-}+\mathrm{UV},\;\mathrm{heat}\;\mathrm{or}\;\mathrm{US}\;\to \;{\mathrm{SO}}_{4}^{•-}+{\mathrm{HO}}^{•}$$
(4)$${\mathrm{HSO}}_{5}^{-}+e^{-}\;\to \;{\mathrm{SO}}_{4}^{•-}+{\mathrm{HO}}^{-}$$
(5)$${\mathrm{HSO}}_{5}^{-}+e^{-}\;\to \;{\mathrm{SO}}_{4}^{2-}+{\mathrm{HO}}^{•}$$
(6)$${\mathrm{HSO}}_{5}^{-}+h^{+}\;\to \;{\mathrm{SO}}_{5}^{•-}+2H^{+}$$

One of the major problems detected in the treatment of WW is the presence of high values of turbidity, total suspended solids, total polyphenols and dissolved organic carbon (DOC) [17,18], which affect the efficiency of the AOPs due to radical scavenging (Equations (7) and (8)) [19]. In a search performed in the Web of Science and Scopus, results showed only 11 works published with the application of sulfate radicals for the treatment of WW. A background for the activation of PMS in real WW has not been established, particularly for WW with high contents of turbidity and TSS.
(7)$$\mathrm{DOC}+{\mathrm{HO}}^{•}\;\to \;{\mathrm{DOC}}_{\mathrm{ox}}^{+}+{\mathrm{HO}}^{-}$$
(8)$$\mathrm{DOC}+{\mathrm{SO}}_{4}^{•-}\;\to \;{\mathrm{DOC}}_{\mathrm{ox}}^{+}+{\mathrm{SO}}_{4}^{2-}$$

Considering this background, the main aim and novelty of this work is the production of a coagulant from almond skin to apply in a CFD process for WW treatment and to recycle the sludge as fertilizer. To decrease more recalcitrant organic matter, the aim is to use a response surface methodology (RSM) Box–Behnken design to define the best SR-AOP to adopt for WW treatment. The influence of pH, temperature, transition metals, single vs. multiple addition of reagents and different radiation sources (UV-A, UV-C and ultrasound) is studied in the removal of organic carbon.

## 2. Material and Methods

### 2.1. Reagents

Potassium peroxymonosulfate (PMS) was acquired from Alfa Aesar (Ward Hill, Massachusetts, US), aluminum sulfate 18-hydrate (10% *w*/*w*, Al_2_(SO_4_)_3_•18H_2_O) was obtained from Scharlau (Barcelona, Spain), cobalt(II) sulfate heptahydrate, iron(II) sulfate heptahydrate (FeSO_4_•7H_2_O) and manganese(II) sulfate monohydrate (MnSO_4_•H_2_O) were acquired from Panreac (Barcelona, Spain), Zinc(II) sulfate heptahydrate was acquired from Merck, (Darmstadt, Germany), copper(II) sulfate 2-hydrate (CuSO_4_•2H_2_O) was acquired from Honeywell Riedel-de-Haën™ (Charlotte, North Caroline, USA), magnesium sulfate heptahydrate (MgSO_4_•7H_2_O), 2,2-diphenyl-1-picrylhydrazyl (DPPH) radical, 6-hydroxy-2,5,7,8-tetramethylchroman-2-carboxylic acid (Trolox), catechin, gallic acid, caffeic acid and sodium chloride (NaCl) were purchased from Sigma-Aldrich (St. Louis, MO, USA). For pH adjustment, sodium hydroxide (NaOH) was used from Labkem (Barcelona, Spain) and sulfuric acid (H_2_SO_4_, 95%) from Scharlau (Barcelona, Spain). Deionized water was used to prepare the respective solutions.

### 2.2. Analytical Determinations

Different physical–chemical parameters were determined to characterize the WW, including turbidity, total suspended solids (TSS), volatile suspended solids (VSS), chemical oxygen demand (COD), biological oxygen demand (BOD_5_), dissolved organic carbon (DOC) and total polyphenols (TPh). The main WW characteristics are shown in Table 1.

The COD analysis was carried out in a COD reactor from Macherey-Nagel (Düren, Germany), and a HACH DR 2400 spectrophotometer (Loveland, CO, USA) was used for colorimetric measurements. The BOD_5_ was determined according to the 5-day BOD test (Standard Method 5210B) using a respirometric OxiTop^®^ IS 12 system (WTW, Yellow Springs, OH, USA). The pH was determined via a 3510 pH meter (Jenway, Cole-Parmer, UK) and conductivity was determined via a portable condutivimeter, VWR C030 (VWR, V. Nova de Gaia, Portugal), in accordance with the methodology of the Standard Methods [20]. The turbidity was determined via a 2100N IS Turbidimeter (Hach, Loveland, CO, USA), and the total suspended solids were measured through spectrophotometry according to Standard Method 2540D using a HACH DR/2400 portable spectrophotometer (Hach, Loveland, CO, USA). The total polyphenols were measured using the Folin–Ciocalteau method, adapted by Singleton and Rossi [21]. UV-vis measurements were performed using a Jasco V-530 UV/VIS spectrophotometer. The total nitrogen (TN) and DOC samples were analyzed via direct injection of the filtered samples into a Shimadzu TOC-L_CSH_ analyzer (Shimadzu, Kyoto, Japan) equipped with an ASI-L autosampler, provided with an NDIR detector, and calibrated with standard solutions of potassium phthalate.

### 2.3. Preparation of Almond Skin Extract (ASE) and Characterization Methodologies

Almond samples (*Prunus dulcis*) from cultivars commonly produced in Trás-os-Montes, northeastern Portugal, were obtained directly from producers located in this region, transported to the laboratory and the skin was removed from the almond. The skin was ground to a fine powder using a 150 W Princess grinder with 2 knife blades (Deluxe, Netherlands). All ground skins were sieved through a 0.4 mm sieve, and the resulting fraction with particle size less than 0.4 mm was used in the coagulation experiments. The extract was prepared by the addition of 12.5 g of prepared powder to 250 mL of a NaCl 1M solution, and the suspension was stirred for 5 h at ambient temperature (298 K) to extract the coagulation-active compounds. Finally, the suspension was allowed to rest for 5 min to sediment the solid parts, and the extract was collected through decantation [22]. The crude extracts were stored in the refrigerator (278 K) and used the following day to avoid aging phenomena and improve reproducibility. The ASE presented a pH of 5.0 ± 0.25.

The chemical spectrum of the almond powder (AP) was obtained via Fourier transform infrared spectroscopy (FTIR), and the KBr sample was analyzed using an IRAffinity-1S Fourier transform infrared spectrometer (Shimadzu, Kyoto, Japan) with the infrared spectrum in transmission mode recorded in the 4000–400 cm^−1^ frequency region. The microstructural characterization of the almond powder was carried out with scanning electron microscopy (FEI QUANTA 400 SEM/ESEM, Fei Quanta, Hillsboro, WA, USA) and the chemical composition was estimated using energy dispersive X-ray spectroscopy (EDS/EDAX, PAN’alytical X’Pert PRO, Davis, CA, USA). The pH of the point of zero charge (pH_PZC_) was determined according to the method described by Oussalah et al. [23], in which 50 mL of 0.01 M NaCl solutions were adjusted to a pH range of 2–12. Then 200 mg of almond skin powder was added to each NaCl solution. The suspensions were stirred for 48 h at room temperature, and the final pH of the solutions (pH_f_) was determined. The pH_PZC_ was obtained from the plot of (pH_f_-pH_i_) versus pHi.

For the preparation of the methanolic extract, 40 mg of AP was weighed and mixed by vortexing with 1 mL of 70% methanol. The mixtures were heated at 70 °C for 30 min and centrifuged at 13,000 rpm at 1 °C for 15 min (Eppendorf Centrifuge 5804 R, Hamburg, Germany). The supernatants were collected and filtered with Spartan filters (0.2 mm) into HPLC amber vials. The methodology of Singleton and Rossi (1965) [21] was used for the quantification of total phenolics in a 96-well microplate. Total phenolics results were expressed as mg gallic acid equivalent (GAE)/g f.w. The total flavonoid content was determined in a 96-well microplate using the colorimetric method described in Dewanto et al. [24]. The total flavonoid content was expressed as mg catechin equivalent (CE)/g f.w. The total ortho-diphenol content was determined in a 96-well microplate, in accordance with the methodology of Soufi et al. [25], and the results were expressed as mg of caffeic acid equivalents/g of dry weight. The 2,2-diphenyl-1-picrylhydrazyl (DPPH) antioxidant activity assay was performed through spectrophotometry, as described by Siddhraju and Becker [26], in a 96-well microplate. The DPPH was expressed as μg trolox equivalent/g f.w.

### 2.4. Coagulation–Flocculation–Decantation Experiments

The coagulation–decantation–flocculation process was performed in a Jar test device (ISCO JF-4, Louisville, KY, USA), with four mechanical agitators powered by a regulated speed engine. The mixing of the ASE with WW samples was performed under a fast mix of 150 rpm/3 min and a slow mix of 20 rpm/20 min, at ambient temperature (298 K). Four different ASE concentrations (0.1, 0.5, 1.0 and 2.0 g/L) were tested against four different pH levels (3.0, 6.0, 9.0 and 11.0), and, after a sedimentation time of 4 h, samples were retrieved for analysis.

### 2.5. Box–Behnken Experimental Design

A Box–Behnken design was employed to assess the effect of different parameters on the UV-A LED SR-AOP treatment of WW1, such as the concentration of PMS (mM, X_1_), the concentration of Co^2+^ (mM, X_2_) and radiation intensity (W m^−2^, X_3_) under fixed conditions (COD = 616 mg O_2_/L, temperature = 323 K, pH = 6.0, time = 120 min). For this study, 15 experiments were performed in triplicate. The levels considered for the Box–Behnken design are listed in Table 2, with three replicates at the center of the design (SR2, SR10 and SR11). Experiments were randomized to maximize the efforts of unexplained variability in the observed response due to external factors.

### 2.6. SR-AOP Set-Up

The experiments were performed on WW2 in a beaker with 500 mL capacity under constant agitation (350 rpm). The temperature of the reaction was controlled with a heating plate (Nahita blue model 692/1, Navarra, Spain) equipped with temperature probe. The following variables were studied: (1) pH (3.0–11.0), (2) temperature (298–363 K), (3) catalyst type (Zn^2+^, Al^3+^, Co^2+^, Cu^2+^, Fe^2+^, Mg^2+^ and Mn^2+^) and (4) reagent addition (single vs. multiple addition). Finally, three radiation sources were used:(1)A UV-A LED system composed of 12 indium gallium nitride (InGaN) LED lamps (Roithner APG2C1-365E LEDs) with a λ_max_ = 365 nm. Each UV-A LED had a nominal consumption of 1.4 W when the current was 350 mA, with an optical power of 135 mW and an opening angle of 120º, making any shadow zone impossible. The radiation was emitted in continuous mode for all 12 UV-A LEDs and was controlled using a power MOSFET in six different current settings, resulting in irradiance levels from 5.2 to 32.7 W m^−2^ measured at a 5 cm distance with a UVA Light Meter (Linshang model LS126A);(2)A Heraeus TNN 15/32 lamp (14.5 cm in length and 2.5 cm in diameter) mounted in the axial position inside the reactor, with 15 W power. The spectral output of the low-pressure mercury vapor lamp emitted mainly (85–90%) at 253.7 nm and about 7–10% at 184.9 nm;(3)An ultrasonic system (Vibracell Ultrasonic processor VCX 500, Sonics & Materials Inc., Danbury, CT, USA) with 500 W power, equipped with a titanium alloy probe (136 mm diameter, 13 mm) and a temperature control probe. For temperature control, a water jacket was installed.

To determine the removal percentage of the parameters, Equation (9) was applied, as follows [27,28]:(9)$$X_{i}\left(\%\right)=\frac{C_{i}\;-{\;C}_{F}}{C_{i}}\text{×}100$$
where $C_{i}$ and $C_{f}$ are the initial concentrations and 100 is the conversion factor.

### 2.7. Phytotoxicity Tests

Phytotoxicity tests were performed via the germination of *Zea mays* (corn) and *Cucumis sativa* (cucumber) seeds (standard species recommended by the US EPA, the US FDA and the OECD [29]). Seeds were immersed in a 10% sodium hypochloride solution for 10 min to ensure surface sterility, then they were soaked in pure water. One piece of filter paper (Whatman filter paper 9 cm, Maidstone, UK) was put into each 100 mm × 15 mm Petri dish, and 5 mL of test medium was added [30]. Seeds were then transferred onto the filter paper, with 10 seeds per dish and a 1 cm or larger distance between each seed. Petri dishes were covered and sealed with tape and placed in a controlled atmosphere with a constant temperature (25 °C), maintained during the course of the experiment with a WTM TS 608-G/2-i (Weilheim, Germany). After 7 days of darkness and 7 days of light, the germination index was determined by Equation (10), in accordance with Varnero et al. [31] and Tiquia and Tam [32], as follows:(10)$$\mathrm{GI}\left(\%\right)=\frac{{\overset{-}{N}}_{SG,T}}{{\overset{-}{N}}_{SG,B}}\text{×}\frac{{\overset{-}{L}}_{R,T}}{{\overset{-}{L}}_{R,B}}\text{×}100$$
where GI is the germination index, ${\overset{-}{N}}_{SG,T}$ is the arithmetic mean of the number of germinated seeds in each extract (wastewater), ${\overset{-}{N}}_{SG,B}$ is the arithmetic mean of the number of germinated seeds in standard solution (distilled water), ${\overset{-}{L}}_{R,T}$ is the mean root length of each extract (wastewater), and ${\overset{-}{L}}_{R,B}$ is the mean root length in control (distilled water). If GI ≤ 50%, then there was a high concentration of phytotoxic substances; if 80% < GI > 50%, then there was a moderate presence of phytotoxic substances; and if GI ≥ 80%, then there were no phytotoxic substances (or they existed in very small dosages).

### 2.8. Statistical Analysis

The coefficients corresponding to the model equation were obtained using Minitab Statistical Software 2018 (State College, PA, USA). All experiments were performed in triplicate, and a one-way ANOVA was carried out to determine any significant differences (*p* < 0.05) using the Tukey test. Results are presented as mean ± standard deviation (SD).

## 3. Results and Discussion

### 3.1. Characterization of Almond Skin Powder

Before the production of the almond skin extract (ASE), it was necessary to characterize the powder obtained from the almond skin. Figure 1a shows the chemical spectrum of the AS by FTIR. The band at 3460 cm^−1^ can be attributed to O-H and N-H stretching vibrations of hydroxyl groups and amide A of polypeptides and amino acids, respectively. The O-H stretching is also associated with the water fraction and with the polyphenol content of the samples [33]. The band at 3190 cm^−1^ is related to the stretching vibration of C=CH cis olefinic groups of unsaturated fatty acids [34]. The bands at 2924 and 2895 cm^−1^ correspond to the symmetric and asymmetric stretching vibrations of aliphatic CH_2_ functional groups, respectively, which are linked with the saturated fatty acid fraction [35]. The band at 1608 cm^−1^ corresponds with the stretching vibration of cis C=C of unsaturated acyl groups [36]. The band at 1508 cm^−1^ corresponds to the bending vibration of the N-H functional group mainly observed in Amide I and Amide II of protein compounds [33]. The bands at 1450 and 1255 cm^−1^ are associated with the presence of CH bending vibrations in CH_2_ and CH_3_, respectively [37]. Finally, a peak at 1016 cm^−1^ is related to the stretching vibration of C-O functional groups characteristic of the carbohydrate fraction [36].

The SEM images (Figure 1b) show that the almond skin powder presents dark spaces that correspond to empty spaces, similar to findings in other works involving plant-based materials [38]. These porous materials allow the adsorption of the NaCl solution and the desorption of material from the powder to the exterior, a necessary characteristic to produce the ASE.

Figure 1c shows major concentrations of potassium (K^+^), calcium (Ca^2+^) and magnesium (Mg^2+^), which are linked to the stress resistance of the plants [39]. The chemical analysis presented in Appendix A shows that AS revealed a total phenolic concentration of 6.51 ± 0.62 mg GAE/g, a flavonoid concentration of 4.30 ± 0.16 mg CE/g, an O-diphenol concentration of 0.65 ± 0.05 mg CAE/g and a DPPH concentration of 8.93 ± 0.34 µg Trolox/g. These results are in agreement with the work of Oliveira et al. [40], who observed similar concentrations in almonds. These results showed that the skin of the almond has antioxidant capacities.

Figure 1d shows the effect of pH on almond skin powder. The pH_PZC_ value determined for the ASP was 4.73. This means that at a pH < 4.73, the surface of the almond skin powder is positively charged, and the biosorption of anionic particles will be favored [41]. However, at a pH > 4.73, the surface of the ASP is negatively charged, increasing the attraction of cations [23].

### 3.2. Coagulation–Flocculation–Decantation Experiments

In this section, almond skin extract (ASE) was applied for the treatment of WW2 via the CFD process. Table 1 shows that WW2 has a complex composition with a high content of organic matter, polyphenols and turbidity. In order to optimize the CFD process and to understand the behavior of ASE, the pH of the wastewater and concentration of ASE were varied. The results showed that the ASE efficiency was affected by the pH of the wastewater. From Figure 2a, results show that a pH of 3.0 achieved a turbidity removal within the range of 94.8 to 96.8% and a TSS removal within the range of 96.9 to 98.1%. This turbidity and TSS removal decreased as the pH increased. On the other hand, the variation in ASE concentration showed little difference regarding turbidity and TSS removal. These results are not in agreement with the work of Hussain Haydar [42], who observed that the natural coagulant *Opuntia stricta* achieved high TSS removal at pH 10.3. Figure 2b shows the influence of ASE on TPh removal. The importance of TPh removal lies in the reduction in the color, which is caused mainly by the polyphenols [27]. The results showed that pH was the main factor that influenced TPh removal, with the highest removal observed at pH 3.0. These results agree with the pH_PHZ_ values, which showed that a pH < 4.73 increased the attraction of the proteins present in the ASE and the negatively charged polyphenols, such as tannins. As the pH increased above the pH_PZC_, the electrostatic repulsion between the proteins in the ASE and the polyphenols increased, explaining the low removal [43]. The COD and DOC removals were studied (Figure 2c), with results showing high removals with 0.5 g/L ASE at pH 3.0 (61.2 and 56.8%, respectively). These results showed that ASE was able to reduce the organic carbon in suspension as well as the dissolved organic carbon. Considering the low levels of organic carbon removed in other works [44], the use of ASE proved to be competitive. Regarding the pH of the WW2 (3.61), the cost of pH changing is avoided, thus this becomes a cheaper process for wastewater treatment.

In order to decrease the environmental impact of sludge generation, this work tested the possibility of recycling the sludge as fertilizer. The recovered sludge was applied as a substrate for cucumber and corn seed germination. The results showed that the application of 1.0 and 2.0 g/L ASE at pH 3.0 and 6.0 could have a toxic effect on seed germination (Figure 3a,b). At the operational conditions selected (0.5 g/L ASE at pH 3.0), the results showed a GI ≥ 80% with good radicular growth; thus, the sludge can be recycled as fertilizer. These results are in agreement with the work of Jorge et al. [9], who observed that the sludge of WW could be recycled as fertilizer if organic coagulants were applied.

### 3.3. SR-AOP Optimization through Response Surface Methodology

In the previous section, ASE showed high efficiency in the removal of turbidity and TSS; however, the coagulant showed limited capacity in the removal of COD. Therefore, in this section, the removal of COD by SR-AOPs was studied. Considering the limited information available regarding the oxidation of organic matter from WW by PMS and Co^2+^, it was necessary to create a model that can help reach the concentration of PMS and Co^2+^ and, at the same time, allow the study of different SR-AOPs. In this section, a Box–Behnken design was applied to achieve the initial conditions of PMS and Co^2+^ concentrations. WW1 was used for this model due to its low COD, thus reducing the reagent requirement. The assessment of the COD removal was performed throughout a range of SR-AOP conditions (n = 15) based on distinct combinations of PMS concentration (X_1_: 0–10 mM), Co^2+^ concentration (X_2_: 0–5 mM) and radiation (X_3_: 0–32.70 W m^−2^). The results of the 15 runs are shown in Table 3. The range of PMS and Co^2+^ concentrations was consistent with those previously assayed in the treatment of agro-industrial wastewaters by other authors [45,46]. The LED lamps were selected as radiation sources due to (1) the LED efficiency, (2) the LED price and (3) the maximum emission wavelength [47,48].

Before optimizing the treatment process, it is necessary to understand which SR-AOP is best fitted to reduce the COD present in the WW. The RSM model allowed the study of different variables to understand how they affect the generation of sulfate radicals (${\mathrm{SO}}_{4}^{•-}$). In Figure 4a, the removal efficiencies of different oxidation systems, which include (1) PMS, (2) UV-A, (3) Co^2+^, (4) Co^2+^ + UV-A, (5) PMS + UV-A, (6) PMS + Co^2+^ and (7) PMS + Co^2+^ + UV-A, can be observed. The reactions involving Co^2+^, PMS, Co^2+^ + UV-A and UV-A reached a COD removal of 7.7, 10.1, 18.3 and 23.1%, respectively. The low removal observed with the application of Co^2+^ and Co^2+^ + UV-A was expected since cobalt alone is not able to generate radicals [49]. The UV-A radiation alone was revealed to be inefficient, which is in agreement with the work of Jorge et al. [18]. The application of PMS alone achieved a reduced COD removal, a result that could be related to the low PMS oxidation potential (1.82 V) [16]. The PMS + UV-A reached higher COD removal than PMS and UV-A alone (33.7%), which is in agreement with the work of Huang et al. [50]. These results could be due to the conversion of the PMS by the UV-A radiation into ${\mathrm{SO}}_{4}^{•-}$ radicals. The highest COD removals were observed with the application of PMS + Co^2+^ and PMS + Co^2+^ + UV-A (44.3 and 62.9%, respectively). The results showed that the activation of PMS by cobalt was higher than the activation by UV-A radiation. It was also observed that a synergy effect occurs with the application of PMS + Co^2+^ + UV-A. This synergy could be linked to the regeneration of Co^3+^ to Co^2+^ by UV-A radiation, which further enhances the conversion of PMS and the generation of ${\mathrm{SO}}_{4}^{•-}$ radicals.

The RSM model also contributes to the optimization of PMS and Co^2+^ concentrations. The increase in PMS concentration up to 5 mM showed an increase in COD removal due to the higher generation of ${\mathrm{SO}}_{4}^{•-}$ radicals. As the PMS concentration increased to 10 mM, results showed a reduction in COD removal (Figure 4b) due to the excess of PMS in the solution, which competed with the organic matter for the ${\mathrm{SO}}_{4}^{•-}$ radicals (Equation (11)) [51]. These results are in agreement with the work of Govindan et al. [52], who observed that high concentrations of PMS led to the consumption of ${\mathrm{SO}}_{4}^{•-}$ radicals, decreasing the degradation of the contaminant pentachlorophenol.
(11)$${\mathrm{HSO}}_{5}^{-}+{\mathrm{SO}}_{4}^{•-}\;\to \;{\mathrm{HSO}}_{4}^{-}+{\mathrm{SO}}_{5}^{•-}$$

The Co^2+^ concentration was observed to have a significant effect on COD removal. Comparing the results obtained in SR1 and SR8, when the PMS:Co^2+^ ratio decreased from 1:0.5 to 1:0.25, the COD removal increased. The excess of Co^2+^ present in SR1 was observed to compete for ${\mathrm{SO}}_{4}^{•-}$ radicals, decreasing the efficiency of the oxidation process (Equation (12)) [53]. These results are in agreement with the work of Rodríguez-Chueca et al. [45], who observed that an excess of cobalt led to the consumption of ${\mathrm{SO}}_{4}^{•-}$ radicals in the treatment of WW.
(12)$$\mathrm{Co}2^{+}+{\mathrm{SO}}_{4}^{•-}\;\to \;\mathrm{Co}3^{+}+{\mathrm{SO}}_{4}^{2-}$$

The quadratic model developed in this work permitted the adjustment of the theoretical values of COD removal to observed values with a low deviation (Table 3), suggesting a successful application of the RSM methodology. From the RSM model, the polynomial equation (Equation (13)) was obtained, and the regression coefficient (R^2^) for this method was 0.931, which means that the model matches the COD removal adequately.
COD = 0.50 + 4.47X_1_ + 7.64X_2_ + 1.018X_3_ − 0.316X_1_X_1_ − 1.234X_2_X_2_ − 0.0118X_3_X_3_ + 0.808X_1_X_2_ − 0.0006X_1_X_3_ − 0.085X_2_X_3_(13)

The regression coefficients of the intercept, linear, quadratic and interaction terms of the model were determined with the application of the least squares method. The effect of linear, quadratic or interaction coefficients on the response was studied via analysis of variance (ANOVA) (Table 4). The degree of significance of each factor is represented by its *p*-value, which indicates that the regression models for COD removal were statistically relevant with a level of significance of *p* = 0.027 < 0.05 (Appendix A). The model did not display a significant lack of fit (*p* > 0.05), with R^2^ = 92.04% (Appendix A); thus, it can be considered a well-fitting model for the described variables. These statistical analyses revealed that the most important variables for the COD removal from the WW were the PMS (X_1_) and Co^2+^ (X_2_) (Table 4). Throughout this statistical model, the most relevant conditions were obtained: [PMS] = 5.88 mM, [Co^2+^] = 5 mM, radiation intensity = 32.7 W m^−2^.

### 3.4. SR-AOPs Applied to a High Load WW

The operational conditions obtained with the RSM were applied in the treatment of WW2, with a COD of 4925 mg O_2_/L. The advantage of creating a model with a WW with a lower COD content is the application of a lower concentration of reagents, decreasing both costs and scavenging reactions. The role of different parameters such as pH, temperature, transition metals, the dosing procedure of reagents and radiation sources were investigated to establish the optimal conditions for the treatment of WW2 with SR-AOPs.

Initially, the pH was varied from 3.0 to 11.0 over 240 min (Figure 5a). The results showed a COD removal of 47.3, 57.0, 59.4 and 56.4%, respectively, for pH levels 3.0, 6.0, 9.0 and 11.0. The low COD removal observed at pH 3.0 was consistent with the non-productive reactions under highly acidic pH levels (Equation (6)), resulting in reduced production of ${\mathrm{SO}}_{4}^{•-}$ radicals, which is in agreement with the work of Huling et al. [54], who observed no considerable differences between pH 3.0 and 6.0 in the degradation of methyl tert-butyl ether (MTBE) by persulfate oxidation. The results showed a favorable range from 6.0 to 9.0, which was in agreement with Yi et al. [55]. Above pH 9.0, a decrease in COD removal was observed. These results were due to the fact that the alkaline solution inhibited the dissolution of Co^2+^ while promoting the complexation and deposition of Co^2+^. In this way, pH 11.0 inhibited the dissolution of Co^2+^, slowing down the activation of PMS and reducing the catalytic activity [13,56]. Considering the initial pH of the WW (4.81) and considering that the difference in COD removal between pH 6.0 and 9.0 was not significant, pH 6.0 is a better choice, since a lower content of NaOH is required to be spent, which is in agreement with the work of Rodríguez-Chueca et al. [46], who observed that pH 6.5 could achieve significant COD removal from WW.

The temperature of the wastewater was observed by other authors to have an influence on the conversion of persulfate and the generation of ${\mathrm{SO}}_{4}^{•-}$ radicals [57,58]. Figure 5b shows the COD removal at different temperatures (298–363 K) using the optimal pH of 6.0. Before the application of the SR-AOPs, it was necessary to understand the effect of temperature without PMS and Co^2+^, so blank experiments were performed. The results in Figure 5b show a maximum COD removal with the application of 363 K (20.6%). Thus, the organic matter appears to be stable at high temperatures. With the application of SR-AOPs, the results showed no significant differences between 298 and 323 K (51.6 and 57.0%, respectively); however, the COD increased significantly with 343 K (78.7%). In the work of Chen et al. [57], it was observed that persulfate was more easily converted in ${\mathrm{SO}}_{4}^{•-}$ radicals at 343 K, which agrees with the results obtained in this work. These results are also in agreement with the works of Rodríguez-Chueca et al. [59] and Jorge et al. [28], who observed that high temperatures are beneficial in activating PMS. Increasing the temperature to 363 K the COD removal is reduced (78.4%), which could suggest that temperatures above 343 K can inactivate the ${\mathrm{SO}}_{4}^{•-}$ radicals, decreasing the efficiency of the reaction. In previous studies, the reduction in organic matter via thermally activated PMS was observed to follow a pseudo first-order kinetic rate [46].

The effect of the nature of the metal catalyst was studied. In this section, seven different sulfate catalysts were tested (ZnSO_4_, Al_2_(SO_4_)_3_, CoSO_4_, CuSO_4_, FeSO_4_, MgSO_4_ and MnSO_4_) to evaluate their effect on the activation of PMS (Figure 5c). The results showed the highest COD removal with catalysts CoSO_4_, FeSO_4_ and ZnSO_4_, with 78.7, 73.0 and 70.8%, respectively. Cobalt has been reported as one of the most effective catalysts for the activation of PMS, promoting a high generation of ${\mathrm{SO}}_{4}^{•-}$ and ${\mathrm{HO}}^{•}$ radicals [55,60]. The regeneration of Co^3+^ to Co^2+^ in the PMS/Co (Equation (2)) is thermodynamically feasible (0.82 V), fast and the process proceeds cyclically many times until PMS is consumed [13]. The application of FeSO_4_ was shown to be as efficient as CoSO_4_. In the work of Latif et al. [61], iron was observed to be effective in the production of ${\mathrm{SO}}_{4}^{•-}$ radicals in the degradation of the organic contaminant bisphenol A (BPA). The ZnSO_4_ catalyst was observed in this work to have a significant effect on the conversion of PMS in ${\mathrm{SO}}_{4}^{•-}$ and ${\mathrm{HO}}^{•}$ radicals, which is in agreement with the work of Fang et al. [62], who observed that zinc could efficiently react with PMS to generate ${\mathrm{SO}}_{4}^{•-}$ and ${\mathrm{HO}}^{•}$ radicals that degrade BPA. Al_2_(SO_4_)_3_ was used as a catalyst in Al^3+^/PMS reaction. The results showed a higher COD removal (68.4%) with the application of catalysts MgSO_4_, CuSO_4_ and MnSO_4_ (68.1, 66.0 and 56.7%, respectively). Aluminum sulfate is highly used in coagulation–flocculation–decantation processes, posing a low risk to the environment with a higher discharge limit (10 mg Al/L) than cobalt (3 mg Co/L) and iron (2 mg Fe/L) [63]. Thus, Al_2_(SO_4_)_3_ can be considered an alternative to cobalt.

To complement this optimization, the dosing procedure of the reagents was investigated, in which the reagents were added in a single step at the beginning of the reaction (S) or five times in a multiple addition (M). Four different ways were then selected to understand the effect of single vs. multiple addition in COD removal: (1) PMS (S) + Co^2+^ (S), (2) PMS (M) + Co^2+^ (S), (3) PMS (S) + Co^2+^ (M) and (4) PMS (M) + Co^2+^ (M) (Figure 5d). The results showed that the application of PMS (S) + Co^2+^ (M) achieved the lowest COD removal (74.2%). These results could be due to a higher concentration of PMS present in solution and insufficient Co^2+^ present to generate the ${\mathrm{SO}}_{4}^{•-}$. In addition, the excess of PMS could have a scavenger effect (Equation (11)), decreasing the efficiency of the reaction, which could explain the decrease in COD removal. The applications of PMS (S) + Co^2+^ (S) and PMS (M) + Co^2+^ (S) showed similar results, with 78.7 and 78.1% COD removal, respectively. These results showed that the catalyst addition had a significant effect on the conversion of PMS and the generation of ${\mathrm{SO}}_{4}^{•-}$ radicals. With the application of PMS (M) + Co^2+^ (M) the COD removal reached 82.3%, the highest of all addition methods. In contrast with PMS (S) + Co^2+^ (S), the multiple dosing of reagents kept the concentrations of PMS and Co^2+^ low in the reactor, suppressing the rate of scavenging reactions and, as a consequence, a more gradual supply of ${\mathrm{SO}}_{4}^{•-}$ radicals were generated, resulting in a more significant COD removal. The influence of multiple dosages of reagents was reported by other authors. For example, in the work of Sun et al. [64], it was observed that the addition of PMS in multiple dosages decreased the self-decomposition of the oxidant caused by the high concentrations of the addition in single mode.

In this work, the COD removal results were well-fitt into a pseudo first-order kinetic model (ln [COD]_t_ = −*k*t + ln [COD]_0_), and the corresponding rate constants were 2.16 × 10^−3^, 2.49 × 10^−3^, 5.82 × 10^−3^ and 6.40 × 10^−3^ min^−1^ for, respectively, 298, 323, 343 and 363 K (Figure 6a). To have a better understanding of the influence of temperature on PMS activation, the Arrhenius equation (ln*k* = lnA − E_a_/RT) was used, and a chart was plotted, fitting ln *k* vs. 1/T (Figure 6b), where A is the frequency factor, $E_{a}$ is the activation energy, R is the universal gas constant and T is the absolute temperature [58]. The results showed a good fit of the data (y = −1901.73X + 0.27647, R^2^ = 0.967). The average $E_{a}$ (16.07 ± 0.58 kJ mol^−1^) was observed to be lower than the activation of PMS reported by Rodriguez-Narvaez et al. [65] (29.9 kJ mol^−1^) in the degradation of a contaminant by PMS/Co^2+^ at a temperature of 343 K. In further experiments, a temperature of 343 K was selected, considering that above this temperature no significant COD removal was observed.

The PMS/Co^2+^/UV system was shown to be effective in the removal of organic matter. In this section, the optimized conditions (pH = 6.0, [PMS] = 5.88 mM, [Co^2+^] = 5 mM, T = 343 K, time = 240 min) were applied to the treatment of WW in three different reactors. A UV-A LED photosystem, a low-pressure UV-C mercury lamp and an ultrasound reactor were used. Figure 7a shows the COD and DOC removal after the application of the optimal PMS/Co^2+^/UV system in the three reactors. Results showed a COD removal of 82.3, 76.0 and 52.2%, respectively, and a DOC removal of 75.8, 64.1 and 38.8%, respectively, for UV-A, UV-C and US. The high mineralization observed with the UV-C reactor can be explained by the photolysis of the PMS, which generates one mole of ${\mathrm{SO}}_{4}^{•-}$ radicals and one mole of ${\mathrm{HO}}^{•}$ radicals per mole of PMS (Equation (3)). In the work of Wang and Chu [66], it was observed that the system PMS/Fe^2+^/UV (254 nm) achieved the highest degradation of 2,4,5-trichlorophenoxyacetic acid, which was attributed to the decomposition of PMS by the UV-C radiation. However, in this work, it was observed that the application of a UV-A reactor achieved higher mineralization than the UV-C reactor. The activation of PMS with UV-A radiation was shown to be effective in the removal of micropollutants from the wastewater [67] and in the removal of organic carbon from WW [46], which is in agreement with the results obtained in this work. Although the photolysis of PMS is negligible with UV-A radiation [68], the application of Co^2+^ appears to be the contributing factor that increased the efficiency of the reaction and higher mineralization of the organic carbon. The activation of PMS by ultrasound was also studied. The US can be used to generate ${\mathrm{SO}}_{4}^{•-}$ and ${\mathrm{HO}}^{•}$ radicals by the decomposition of PMS (Equation (3)) [69]. The use of ultrasound radiation showed a high removal of organic carbon, although the results were shown to be lower than those of UV-A and UV-C radiation. These results are in agreement with the work of Lu et al. [70], who observed a 71.4% atrazine degradation in the PMS/US system.

Among the organic compounds that are present in the WW, polyphenols are linked to its toxicity, since aromatic compounds are difficult to degrade by microorganisms. Polyphenols have a significant contribution to the dark color of the WW, which limits the penetration of UV radiation and the catalyst regeneration [71]. In this section, the TPh removal was evaluated as a function of the three reactors, with results showing a near complete removal after the 240 min of reaction (Figure 7b). These results are linked with the generation of ${\mathrm{SO}}_{4}^{•-}$ and ${\mathrm{HO}}^{•}$ radicals by the three systems, which are extremely powerful, non-selective and capable of oxidizing most organic compounds. The ${\mathrm{SO}}_{4}^{•-}$ and ${\mathrm{HO}}^{•}$ radicals are capable of attacking the phenol rings of phenolic compounds, yielding benzoic and cinnamic acids, flavonoids and anthocyanins, and then the rings of these compounds break up to give organic acids and finally CO_2_ [28,72].

One of the main factors that affected the efficiency of organic carbon removal was the pH of the wastewater. Therefore, the pH of the reaction was monitored during the 240 min. The results in Figure 7c show that all three systems had a large drop in pH within the first 15 min, which could be linked to the production of ${\mathrm{SO}}_{4}^{•-}$ and ${\mathrm{HO}}^{•}$ radicals. This fact was previously reported by Esteves et al. [73], who observed a high decrease in the pH in the treatment of high-strength olive mill wastewater using a Fenton-like oxidation process. The follow-up of the pH is important because the type and rate of the produced radicals are among the important effects of pH changes. At pH 6.0, the ${\mathrm{SO}}_{4}^{•-}$ radicals were predominant [74,75]. As the reaction unfolded, a significant reduction in the pH took place, caused by the hydrolysis of the ${\mathrm{SO}}_{4}^{•-}$ and the cobalt ions. Therefore, it could be perceived that a higher concentration of ${\mathrm{SO}}_{4}^{•-}$ radicals leads to a more significant pH decline, which is in agreement with the work of Li et al. [76].

### 3.5. CFD + PMS/Co^2+^/Radiation

In the previous section the legal limits for wastewater discharge in a sewage network were not achieved for the UV-C and US reactors (COD < 1000 mg O_2_/L); therefore, in this section, the CFD + PMS/Co^2+^/radiation combined treatment was studied. The CFD process was performed as a pre-treatment followed by oxidation with SR-AOPs. Figure 8a shows the evolution of COD and DOC removal with the combined system. Results showed a COD removal of 85.9, 82.6 and 80.2% and a DOC removal of 83.3, 79.1 and 74.5%, respectively, for CFD/PMS/Co^2+^/UV-A, CFD/PMS/Co^2+^/UV-C and CFD/PMS/Co^2+^/US. Two factors were observed to increase the reactions’ efficiency: (1) the removal of turbidity, TSS and TPh by the ASE, which clarified the wastewater, allowing better penetration of the radiation, and (2) the removal of the organic matter in suspension, which acted as a scavenger of ${\mathrm{HO}}^{•}$ and ${\mathrm{SO}}_{4}^{•-}$ radicals. These results are in agreement with the work of Jaafarzadeh et al. [77], who observed that the application of CFD as a pre-treatment increased the efficiency of electro-activated ${\mathrm{HSO}}_{5}^{-}$ to treat pulp and paper wastewater. In comparison to other works, such as Amor et al. [78], results showed that the application of the CFD/SR-AOP system achieved higher organic matter removal from WW with lower reagent consumption.

In order to evaluate the mineralization capacity of the different processes, an efficient parameter was applied, as a partial oxidation efficiency (μ_partox_), which can be determined by Equations (14) and (15) [79], as follows:(14)$${\mathrm{COD}}_{\mathrm{partox}}=(\frac{{\mathrm{COD}}_{0}}{{\mathrm{DOC}}_{0}}-\frac{\mathrm{COD}}{\mathrm{DOC}})\times \mathrm{DOC}$$
(15)$$\mu_{\mathrm{partox}}=\frac{{\mathrm{COD}}_{\mathrm{partox}}}{{\mathrm{COD}}_{0}-{\mathrm{COD}}_{t}}$$
where µ_partox_ is the partial oxidation efficiency that is between 0 and 1. The µ_partox_ is 0 when only total oxidation occurs, and the value of 1 represents the ideal condition in which only partial oxidation occurs. In fact, total oxidation occurs in µ_partox_ < 0.5, while partial oxidation is dominant in µ_partox_ > 0.5. Figure 8b shows that with the application of UV-A and UV-C reactors, the reduction in organic carbon predominantly occurred via total oxidation reactions, becoming more pronounced at prolonged times, similar to the work of Papastefanakis et al. [80]. With application of the PMS/Co^2+^/US and CFD/PMS/Co^2+^/US systems, partial oxidation reactions occurred at 15 and 30 min, respectively, following total oxidation reactions until 240 min of reaction. These results can be explained by the high reduction in the organic matter by the ${\mathrm{HO}}^{•}$ and ${\mathrm{SO}}_{4}^{•-}$ radicals, which increased the mineralization.

In Figure 8c, the effect of the combined treatment in the removal of TPh can be observed. The ASE was responsible for the removal of 83.2% of the TPh content from the WW, decreasing the dark color of the WW and allowing a better penetration of the radiation. With the application of the UV-A, UV-C and US reactors, the TPh removal rate increased significantly. These results were in agreement with the work of Jorge et al. [18], who observed that application of the CFD process boosted the TPh removal by the SR-AOPs.

Figure 8d shows the impact of the different treatment processes in the BOD_5_ removal and, consequently, in the biodegradability. The applications of the PMS/Co^2+^/UV-A, PMS/Co^2+^/UV-C and PMS/Co^2+^/US achieved a BOD_5_ removal of 69.6, 63.5 and 44.4%, respectively. With the application of the CFD process, the BOD_5_ removal was increased to 73.9, 68.7 and 67.0%, respectively. The biodegradability was also evaluated, with results showing an increase from 0.31 (CFD process) to 0.54, 0.53 and 0.49, respectively. The combination of processes, was able to remove part of the recalcitrant matter, increasing the biodegradable fraction, and allowing a subsequent biological treatment. These results were in agreement with the works of Amor et al. [81] and Rodríguez-Chueca et al. [82], who observed that the combined photo-Fenton and CFD processes increased the biodegradability of crystallized-fruit wastewater.

## 4. Conclusions

Wine and almond production are two major Portuguese agro-industries with enormous weight in Portugal’s economy. The WW generated from wine production is of environmental concern due to the high content of organic matter and polyphenols. From almond production, the skin, often neglected by the food industry, is used in this work to produce an almond skin extract (ASE). The results show that ASE achieves the highest results regarding turbidity, TSS, TPh and COD removal with the application of 0.5 g/L ASE at pH 3.0. It is concluded that the pH has a considerable effect on the ASE efficiency, considering the isoelectric point (4.73). The sludge generated by the treatment with ASE can be recycled as fertilizer, allowing its valorization. The response surface methodology (RSM) associated with a Box–Behnken design was revealed to be one of the most appropriate methods for the optimization of the basic conditions (PMS and Co^2+^ concentration and radiation) for COD removal from WW. The efficiency of the SR-AOP is concluded to be dependent on factors, such as pH, temperature, type of transition metal and manner of addition. Under the optimal conditions, pH = 6.0, [PMS] = 5.88 mM, [Co^2+^] = 5.0 mM, T = 343 K, reaction time = 240 min, a COD removal was achieved of 82.3, 76.0 and 52.2%, respectively, for UV-A, UV-C and US reactors. The combination of CFD with SR-AOPs has a synergic effect, in which the ASE removes 61.2% of COD and the combined ASE/reactors remove 85.9, 82.6 and 80.2%, respectively, with UV-A, UV-C and US. Moreover, the combination of processes allows all reactors to achieve the Portuguese legal value of COD (≤1000 mg O_2_/L) for wastewater to be discharged as municipal wastewater in a wastewater treatment plant (WWTP).

## Figures and Tables

**Figure 1 ijerph-20-02486-f001:**
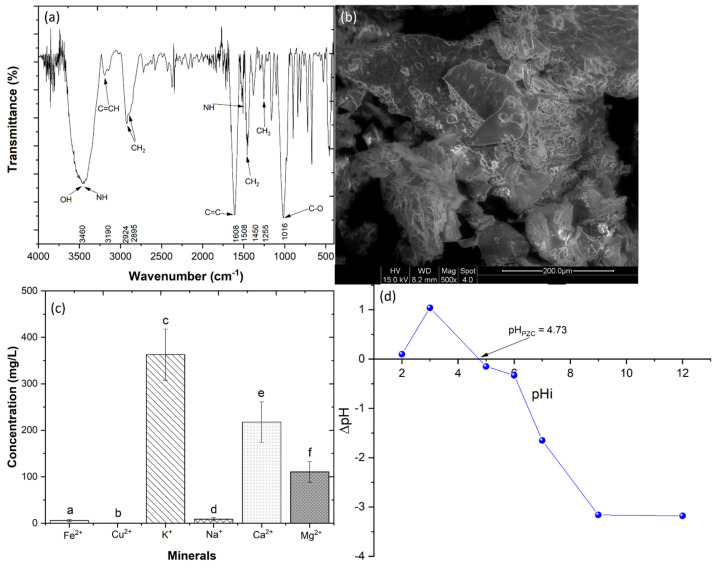
Almond skin powder characterization: (**a**) FTIR spectrum, (**b**) SEM image (500×), (**c**) mineral analysis and (**d**) pH_PZC_ of almond powder. The pHi is the initial pH value of the solution, and ΔpH is the difference between final pH values (after contact with the AP) and pHi values. Means in bars with different letters represent significant differences (*p* < 0.05) between different minerals.

**Figure 2 ijerph-20-02486-f002:**
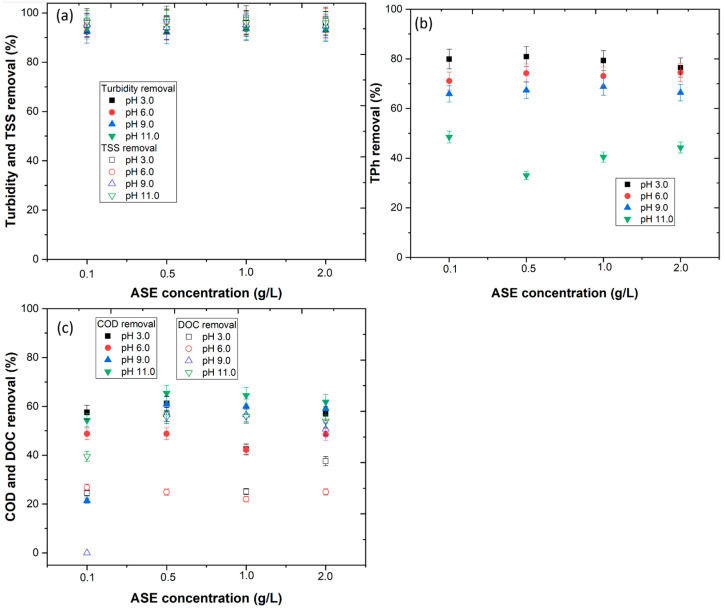
Effect of ASE concentration (g/L) vs. pH in (**a**) turbidity and TSS removal, (**b**) TPh removal, and (**c**) COD and DOC removal from winery wastewater. Operational conditions: fast mix (rpm/min) = 150/3, slow mix (rpm/min) = 20/20, sedimentation time of 4 h.

**Figure 3 ijerph-20-02486-f003:**
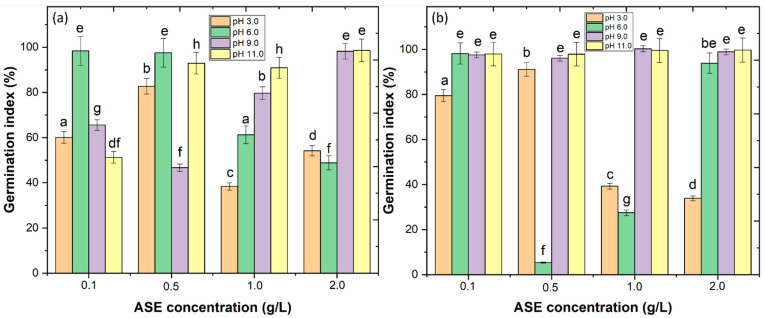
Effect of ASE concentration vs. pH in (**a**) germination index of cucumber and (**b**) corn seeds. Means in bars with different letters represent significant differences (*p* < 0.05) within each ASE concentration by comparing the pH.

**Figure 4 ijerph-20-02486-f004:**
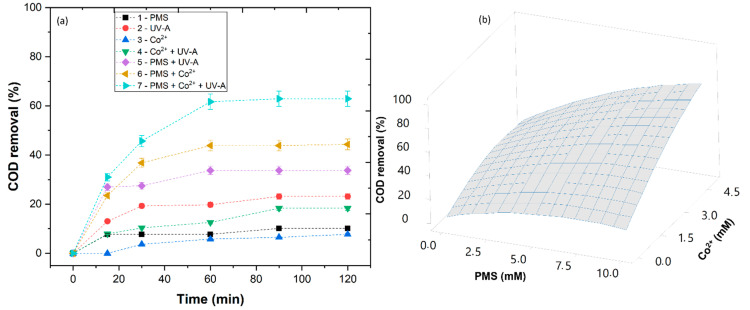
SR-AOP optimization: (**a**) experiments with different oxidation systems in the reduction in COD from WW1, (**b**) response surface methodology. Operational conditions: COD = 616 mg O_2_/L, pH = 6.0, [PMS] = 5 mM, [Co^2+^] = 2.5 mM, temperature = 323 K, radiation UV-A 32.7 W m^−2^, reaction time = 120 min.

**Figure 5 ijerph-20-02486-f005:**
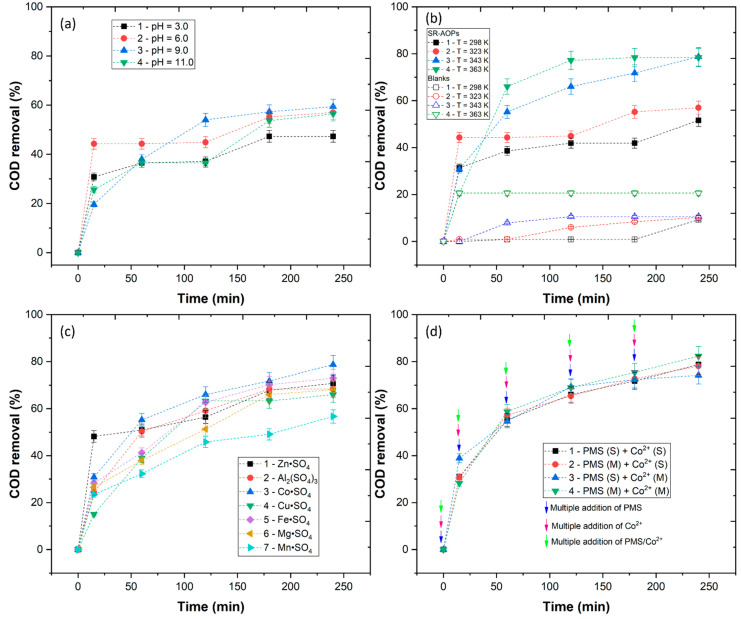
COD removal in the optimization of (**a**) pH ([PMS] = 5.88 mM, [Co^2+^] = 5 mM, radiation UV-A, T = 323 K), (**b**) temperature (SR-AOPs: pH = 6.0, [PMS] = 5.88 mM, [Co^2+^] = 5 mM, radiation UV-A; blanks: pH = 6.0, radiation UV-A), (**c**) type of metal catalyst (pH = 6.0, [PMS] = 5.88 mM, [*M*^n+^] = 5 mM, radiation UV-A, T = 343 K) and (**d**) dosing of PMS and Co^2+^. S—single addition, M—multiple addition (pH = 6.0, [PMS] = 5.88 mM, [Co^2+^] = 5 mM, radiation UV-A, T = 343 K).

**Figure 6 ijerph-20-02486-f006:**
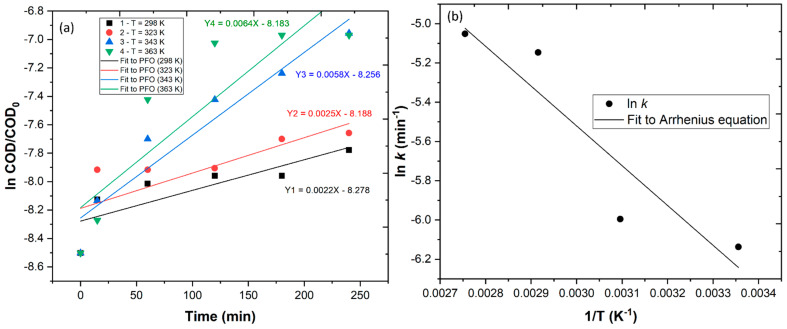
(**a**) Pseudo first-order of COD removal at different temperatures and (**b**) plot of ln *k* vs. 1/T for $E_{a}$ estimation using the Arrhenius equation.

**Figure 7 ijerph-20-02486-f007:**
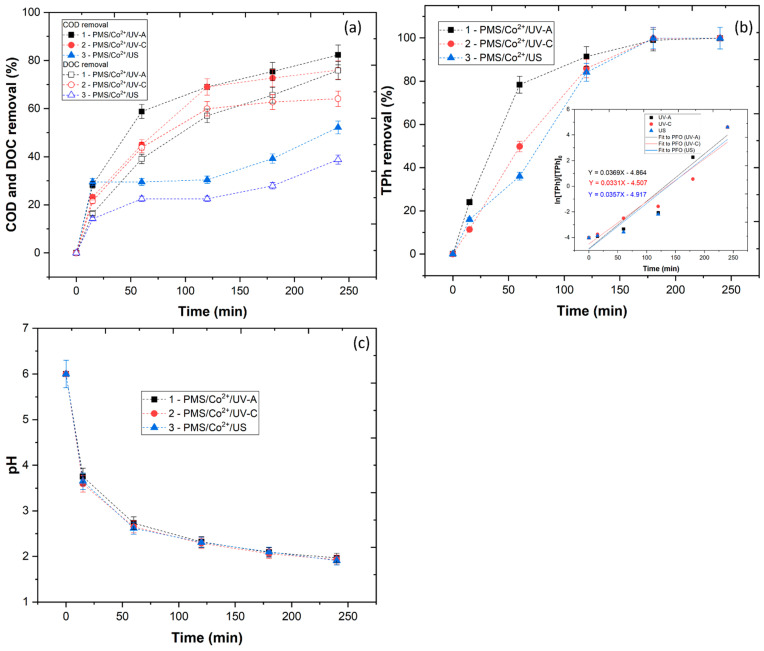
Evaluation of the PMS/Co^2+^/UV system with three radiation reactors (UV-A, UV-C and US) in (**a**) COD and DOC removal, (**b**) TPh removal and (**c**) pH. Operational conditions: pH_i_ = 6.0, [PMS] = 5.88 mM, [Co^2+^] = 5 mM, T = 343 K, time = 240 min.

**Figure 8 ijerph-20-02486-f008:**
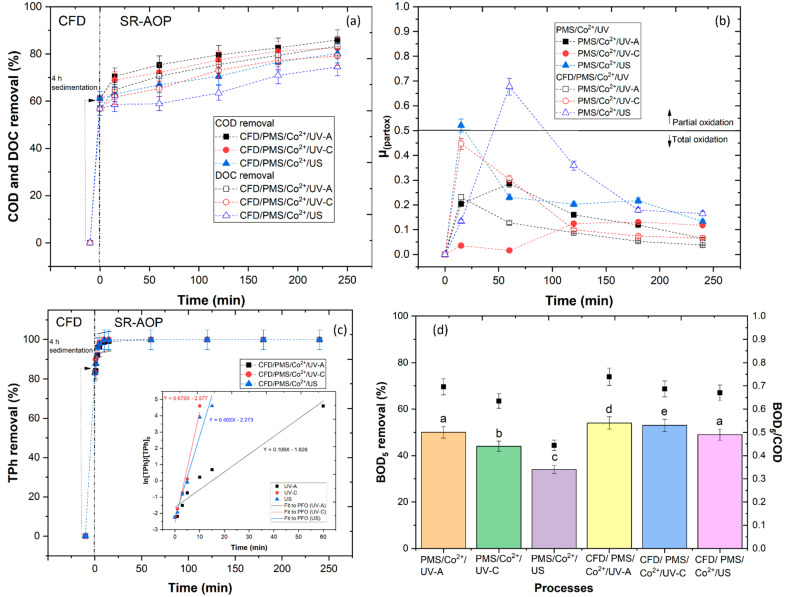
Effect of CFD/PMS/Co^2+^/radiation in (**a**) COD and DOC removal, (**b**) μ_(partox)_, (**c**) TPh removal and (**d**) BOD_5_ removal and BOD_5_/COD. Means in bars with different letters represent significant differences (*p* < 0.05) within BOD_5_/COD by comparing treatment processes. CFD operational conditions: pH = 3.0, [ASE] = 0.5 g/L, fast mix (rpm/min) = 150/3, slow mix (rpm/min) = 20/20, sedimentation time 4 h. SR-AOP operational conditions: pH = 6.0, [PMS] = 5.88 mM, [Co^2+^] = 5 mM, radiation UV-A 32.7 W m^−2^, UV-C 15 W, US 500 W, T = 343 K, reaction time = 240 min.

**Table 1 ijerph-20-02486-t001:** Physicochemical characteristics of winery wastewater (mean ± SD).

Parameter	Values	
	WW1	WW2
pH (Sorensen scale)	3.95 ± 0.20	3.61 ± 0.24
Conductivity (µS cm^−1^)	45 ± 2.3	285 ± 14.3
Turbidity (NTU)	69 ± 4	649 ± 32
Total suspended solids—TSS (mg L^−1^)	200 ± 10	1405 ± 70
Dissolved organic carbon—DOC (mg C L^−1^)	138 ± 7	976 ± 49
Total nitrogen—TN (mg N L^−1^)	3.4 ± 0.2	10.7 ± 0.5
Chemical oxygen demand—COD (mg O_2_ L^−1^)	616 ± 31	4925 ± 246
Biochemical oxygen demand—BOD_5_ (mg O_2_ L^−1^)	163 ± 8	1438 ± 72
Biodegradability—BOD_5_/COD	0.26 ± 0.01	0.29 ± 0.02
Total polyphenols—TPh (mg gallic acid L^−1^)	1.90 ± 0.1	49.5 ± 2.5
Absorbance at 254 nm (diluted 1:25)	0.102 ± 0.005	0.198 ± 0.010
Absorbance at 254 nm (diluted 1:10)	0.124 ± 0.006	0.356 ± 0.018

**Table 2 ijerph-20-02486-t002:** Symbols and coded factor levels for the considered variables.

Independent Variables	Code	Levels		
		−1	0	1
[PMS] (mM)	X_1_	0	5	10
[Co^2+^] (mM)	X_2_	0	2.5	5.0
Radiation	X_3_	0	16.35	32.70

**Table 3 ijerph-20-02486-t003:** Box–Behnken design: effect of operational variables on COD removal yield ([PMS] = 0–10 mM; [Co^2+^] = 0–5 mM; radiation = 0–32.70 W m^−2^). Operational conditions: COD = 616 mg O_2_/L, temperature = 323 K, pH = 6.0, reaction time = 120 min.

Experiments	Coded Level			Response Values	
	[PMS]	[Co^2+^]	Radiation	COD Removal (%)	
	(mM)	(mM)		Observed	Predicted
SR1	5	5.0	32.70	54	49.1
SR2	5	2.5	16.35	46.4	46.4
SR3	0	5.0	16.35	16.8	14.4
SR4	5	5.0	0.00	44.3	42.5
SR5	10	5.0	16.35	58.7	67.8
SR6	5	0.0	32.70	33.7	35.5
SR7	0	0.0	16.35	23.1	14.0
SR8	10	2.5	32.70	62.9	58.7
SR9	10	2.5	0.00	52.5	45.2
SR10	5	2.5	16.35	46.4	46.4
SR11	5	2.5	16.35	46.4	46.4
SR12	5	0.0	0.00	10.1	14.9
SR13	10	0.0	16.35	24.6	27.0
SR14	0	2.5	32.70	18.3	25.6
SR15	0	2.5	0.00	7.7	11.9

**Table 4 ijerph-20-02486-t004:** *F* and *p*-values for selected responses for each obtained coefficient.

Variable	X_1_	X_2_	X_3_	X_1_X_1_	X_1_X_2_	X_1_X_3_	X_2_X_2_	X_2_X_3_	X_3_X_3_
*F*-value	29.59	11.36	4.95	3.08	5.48	0.00	2.95	0.65	0.50
*p*-value	*	*	n.s.	n.s.	n.s.	n.s.	n.s.	n.s.	n.s.

X_1_: PMS (mM); X_2_: Co^2+^ (mM); X_3_: radiation. n.s.: non-significant. Significant at * *p* < 0.05.

## Data Availability

Not applicable.

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
