# Peer review of "Treatment of Winery Wastewater by Combined Almond Skin Coagulant and Sulfate Radicals: Assessment of Activators"

_ijerph, 2023, doi:10.3390/ijerph20032486_

Round 1
Reviewer 1 Report
Comments on the manuscript entitled “Treatment of winery wastewater by combined almond skin coagulant and sulfate radicals: assessment of HSO5- activators” authored by N. Jorge et al. (ijerph-2163210)
The manuscript reports the application of coagulation-flocculation-decantation (CFD) process with an organic coagulant based in almond skin extract to winery wastewater treatment. Another subject is the treatment of organic recalcitrant matter by sulfate radical advanced oxidation processes (SR-AOPs) using a response surface methodology (RSM) Box-Behnken design and evaluation the efficiency of UV-A, UV-C and ultrasound (US) reactors.
From a scientific point of view, the manuscript is very interesting as it allows to study how is the influence of pH, temperature, transition metals, single vs multiple addition of reagents and different radiation sources (UV-A, UV-C and ultrasound) in the removal of organic carbon.
Under my point of view, the manuscript deserves publication in International Journal of Environmental Research and Public Health after minor revision, provided that the following changes are made to improve its quality:
1. In the Abstract, I think in the objective 4 are studied the efficiency of the three different reactors but not particularly the energy efficiency and operational costs: “(4) evaluate the energy efficiency and operational cost of the UV-A, UV-C and ultrasound (US) reactors.” Why you refer in the Abstract the “energy efficiency and operational costs”?
2. Why authors did two optimizations? Wouldn't one be enough?
3. Why were two different winery effluents used (WW1 and WW2)? Couldn't just have used one as a study example?
4. Phytotoxicity tests were carried out with corn and cucumber as target species. Why were these two species chosen?
5. Please check the whole manuscript in order to correct some English grammar mistakes.
Reviewer 2 Report
The manuscript entitled ‘Treatment of winery wastewater by combined almond skin coagulant and sulfate radicals: assessment of HSO5 - activators’ investigated a method of coagulation-flocculation-decantation (CFD) process with an organic coagulant based in almond skin extract (ASE) for a for WW treatment and subproducts valorization. Also, investigation of the sludge recycling as fertilizer, treatment of the organic recalcitrant matter by sulfate radical advanced oxidation processes (SR-AOPs) and evaluating of the energy efficiency and operational cost of the UV-A, UV-C, and ultrasound (US) reactor were performed.
Overall, this is a good study, with useful information achieved; worthy of publication. The authors carried out a good number of experiments with high removal efficiencies and reported an appreciable amount of data. Figures and Tables have effective data presentation and make it easier for readers to understand research data.
However, in its current version, the manuscript presents many weaknesses.
1. Abstract is too long, with many numbers. Please revise, and keep only the most essential info. The abstract contains more than 350 words; I suggest to be reduced to about 250 words to be easier for the reader.
2. Graphical abstract. It is a bit confusing, too much information. I suggest to be improved by a graphic designer.
Consequently, this work could be accepted by the International Journal of Environmental Research and Public Health after a minor revision.
Reviewer 3 Report
Good article but some major changes are required.
Methods-
In some instances, methods aren't fully reproducible.
Line 142 what type of mill was used- miac discs, hammer mill etc more details are required
Line 145 what temperature was this carried out at?
Line 147 how long were they stored before being used? was everything stored for the same length of time?
Line 184- why box-Behnken over CCD of factorial designs ? what was the reason for this choice?
Line 189 - where only these 3 samples ran in triplicate- the whole design should be run in triplicate taking into account block on reps if it all cannot be completed in one go. Please give more detail on this
Line 223- why was only one wash of sodium hypochlorite done- was a control done with no washing ? as sodium hypochlorite is known to have stimulatory effects on seeds during germination? what is the germination potential without washing ?
Line 242- why was the ANOVA carried out on separate software when running DOE-RSM in minitab allows for this analysis to occur?
What samples had ANOVA carried out - what type of ANOVA - one way ??? was post- hoc test done ? this need a major revison
General methods comments
It is not indicated if any experiment was run in triplicate, I have serious concerns about this for the reproducibility of tha data!
Results
Line 269-275 where is this data present - what table or figure ?
Figure 1C - why no error bar on fe or Na . why ? what do the letters mean no explanation is given?
Table 1 is the plus/minus figure standard deviation or error no information is given?
Figure 2- the legends in the middle of the graphs are unsuitable- in COD and DOC removal at 1.5g/L no points can be seen ! same for ASE. this needs to be revised. Again what are the error bars showing?
Figure 3- A lot of the error bars are the same size- why is this? what do they represent? the legend should be taken out of the main graph area. how were p values detemine for letters? as no mention of a post hoc test in methods
RSM section 3.3
This section is good however results are missing what was the p values of the overall model, the lack of fit values etc- without giving these results, it cannon be determined if this is an adequate model for this system. This need to be given for the data set in the paper.
N=15, this indicated this experiment was not run in triplicate and therefore the results cant be reviewed correctly, as one can argue that the experiment is not reproducible! it also shows that the methods that indicate that set runs were done in triplicate were in fact not as n=18 if that was the case.
Likewise in table 3 if this was done in triplicate standard deviation should be given
Line 365- how can this be said when no data for the model for stats is given bar the coefficients p-value and f values and at that no p values number are given
Figure 4 b- what was this made on ? can the surface plot have colours? It does not look like a Minitab output. again what do the error bars represent?
All results need major work - error bars all look very similar- no indication of what post hoc testing was done . remove legands form main graph area
Reviewer 4 Report
Reviewer comments
Dear Editor (s),
I am so grateful for giving me the privilege to review the paper entitled “Treatment of winery wastewater by combined almond skin coagulant and sulfate radicals: assessment of HSO5- activators” by Nuno Jorge and Co-authors.
After a peer review of the manuscript, I only found a few adjustments in the “Results and discussion” section that need to be attended to for better quality of the paper.
I hereby confirm that once these minor comments mentioned below have been fully attended to by the authors, the amended draft can be published in the journal of “International Journal of Environmental Research and Public Health” under your supervision.
Comments and suggestions
Abstract
The abstract is well written and covers all aspects discussed in the manuscript.
Introduction
Well written and is a high quality for publication.
Materials and Methods
No comments
Results and conclusion
1. In section 3.1. Characterization of almond skin powder, the word “…associated...” is used 7 times at least. Can the authors try to use synonym terminologies to avoid such repetition.
2.In L346, just below Table 3, in the following “…degrade the COD…” the word “…degrade…”is not appropriate and should be replaced by terminologies such as “reduce…”.
3.Similar comment is applicable to the word “…degradation…” in the caption of Figure 4. “…degradation…” should be removed and replaced with “reduction”.
Round 2
Reviewer 3 Report
n/a